# Hydrochemical Characteristic of Groundwater and Its Impact on Crop Yields in the Baojixia Irrigation Area, China

**Wenwen Feng** [1,2]**, Hui Qian** [1,2,*]**, Panpan Xu** [1,2] **and Kai Hou** [1,2]

1   School of water and Environment, Chang'an University, Xi'an 710054, China; fww@chd.edu.cn (W.F.); xupanpan0212@163.com (P.X.); houkai19911219@163.com (K.H.)
2   Key Laboratory of Subsurface Hydrology and Ecological Effect in Arid Region of Ministry of Education, Chang'an University, Xi'an 710054, China
*   Correspondence: qianhui@chd.edu.cn; Tel.: +86-029-82339327

**Abstract:** While irrigated crops produce much higher yields than rain-fed crops, the ionic components of irrigation water have important effects on crop yield. Groundwater is widely used for irrigation in the Baojixia irrigation area in China. The chemical characteristics and water quality of groundwater in the Baojixia irrigation area were analyzed and evaluated to study the impact of groundwater quality on crop yield. Results showed cations in the groundwater to mainly be $Na^+$, $Ca^{2+}$, and $Mg^{2+}$, whereas the anions are mainly $HCO_3^-$, $SO_4^{2-}$, and $Cl^-$. Water-rock interaction and cation exchange were identified as the main factors affecting hydrogeochemical properties from west to east. The study found salinity and alkalinity of groundwater in the western region of the study area to be low, and therefore suitable for irrigation. Groundwater in the eastern part of the study area was found to have a medium to high salinity and alkalinity, and is therefore not recommended for long-term irrigation. The groundwater irrigated cultivation of wheat and corn in the research area over 2019, for example, would have resulted in a drop in the annual crop output and an economic loss of 0.489 tons and $0.741 \times 10^4$ yuan, respectively. Irrigation using groundwater was calculated to result in the cumulative loss of crop yields and an economic loss of 49.17 tons and $80.781 \times 10^4$ yuan, respectively, by 2119. Deterioration of groundwater quality will reduce crop yields. It is recommended that crop yields in the study area be increased by strengthening irrigation water management and improving groundwater quality.

**Keywords:** irrigation quality; hydrogeochemical; sodium absorption ratio (SAR); crop yield; food security

## 1. Introduction

Human activities and natural environmental changes are the two main driving forces of regional hydrology and changes to water resources [1–4]. Population growth and the acceleration of industrialization have led to excessive demand for natural resources, which has subsequently resulted in many social and ecological problems. These issues inevitably exacerbate the human impact on the environment, particularly on surface water and groundwater resources [5–9]. Groundwater is the most reliable source of water for human survival, particularly in arid and semi-arid regions such as northwest China, where precipitation and surface runoff are scarce and large volumes of groundwater are extracted for domestic, agricultural, and industrial activities. The long-term use of river water for irrigation results in raised groundwater levels, increased soil salinity, and deterioration of groundwater quality [1,10–13]. The yield of wheat and corn depends on the quality of irrigation water and soil. Soil quality is affected by many factors such as soil type, drainage method, irrigation method, fertilizer,

and moisture. It is very difficult to study. Therefore, research on irrigation water quality is nevertheless of great significance for long-term management planning of crop yield [9,14–17].

The Maas-Hoffman model [18] was the earliest model-derived to study the relationship between irrigation water quality and crop yield. The model sets a threshold salt limit parameter to describe the salt tolerance potential of crops. Past studies have shown that when soil salt content is higher than the threshold, the crop yield decreases with increasing soil salt content [3,14,19,20]. Different growth stages of plants have different thresholds and salt limits. Previous studies have found that the seedling stage is more sensitive to salts compared to the mature stage [14,15,21]. Also, the growth of the plant is dependent on foliage receiving sufficient sunlight. Salinity can have multiple detrimental effects on plant growth. Too much salt in soil inhibits water absorption by plants, resulting in physiological drought. Too many salt ions in plants, such as $Na^+$, inhibits the absorption of other necessary ions and affects the normal production and development of plants [6,19,21,22]. The salinity and sodium absorption ratio (SAR) of irrigation water is used to assess the impact of soil on crop yield [8,15,17,23]. Therefore, the evaluation of irrigation water quality is of great significance for the study of plant growth.

The Baojixia irrigation area is currently the largest artesian diversion irrigation area in Shaanxi Province, China. Irrigation water is sourced from the Weihe River and groundwater. In recent years, increases in water consumption in the upper Weihe River and the continuous deterioration of the ecological environment have led to decreased river flow and insufficient surface water resources [10,24]. During the dry season in particular the river receives little flows, or sometimes none. This results in a large supply-demand deficit in the irrigation districts, and farmers increasingly account for the shortfall by using groundwater [14,25]. The economy and population of the region have grown sharply with the establishment of the Guanzhong-Tianshui Economic Zone as a national development zone. In the region, agriculture is regarded as fundamental for economic and social development. Therefore, the further elucidation of the impact of groundwater quality on crop yield is of important practical significance for ensuring food security in the region [10,25,26]. The interactions between groundwater aquifer minerals have important effects on hydrogeochemistry. Gibbs [27] and Piper [28] diagrams are often used to describe hydrogeochemical processes and their changes. Principal component analysis (PCA) has been used to identify the many factors that affect the chemical characteristics of groundwater. The purpose of PCA in this context is to analyze the main hydrogeochemical processes and their evolution [29,30]. Therefore, methods such as charts, statistical analysis and geochemical simulations are used to evaluate groundwater quality and to identify the evolution of hydrogeochemical processes. At the same time, the relative yield formula established by the Mass-Hoffman model has been used to study the effect of groundwater quality on crop yield. Finally, the economic and social impacts of crop reductions were analyzed. The purpose of the present study was to quantitatively assess the impacts of groundwater quality on crop yields and to translate these impacts into economic values. It is hoped that the present study will increase attention on the quality of groundwater for irrigation and will strengthen the management of agricultural water use, improve water quality, maintain food production, and ensure sustainable and healthy economic and social development.

## 2. Study Area

### 2.1. Location and Climate

The Baojixia irrigation area is located in the central of Shaanxi Province, China (Figure 1). The study area is located at 106°51′–108°48′ E, 34°09′–34°44′ N, north of the Weihe River and south of the North Mountains, and has an area of 2355 km². The northern part of the study area falls within the piedmont alluvial fan and loess terrace area, whereas the southern region is located in the Wei River terrace. This area contains important grain farms and is also the industrial base in Shaanxi Province. In the study area, the total cultivated land area is 1954 km², of which the area that can be irrigated by water is 1890 km². Generally, the coefficient of agricultural water shortage is used to express the ratio of crop water demand under component irrigation to actual irrigation water in agriculture. In the study area,

the coefficient of the degree of agricultural water shortage is 1.19, mostly in water-scarce areas [24,26]. Approximately 63.19% and 55.24% of total surface water and groundwater are used in the irrigation area, respectively [31]. The method of agricultural irrigation remains dominated by flooding, which is recognized to be inefficient. Research has shown that a deficit in the water supply-demand balance from 1991 to 2000. The average, largest and smallest annual water deficits in the study area to date have been 143 million m$^3$, 463 million, and 119 million m$^3$, respectively. The degree coefficient of the annual average water shortage is 39.9% [31].

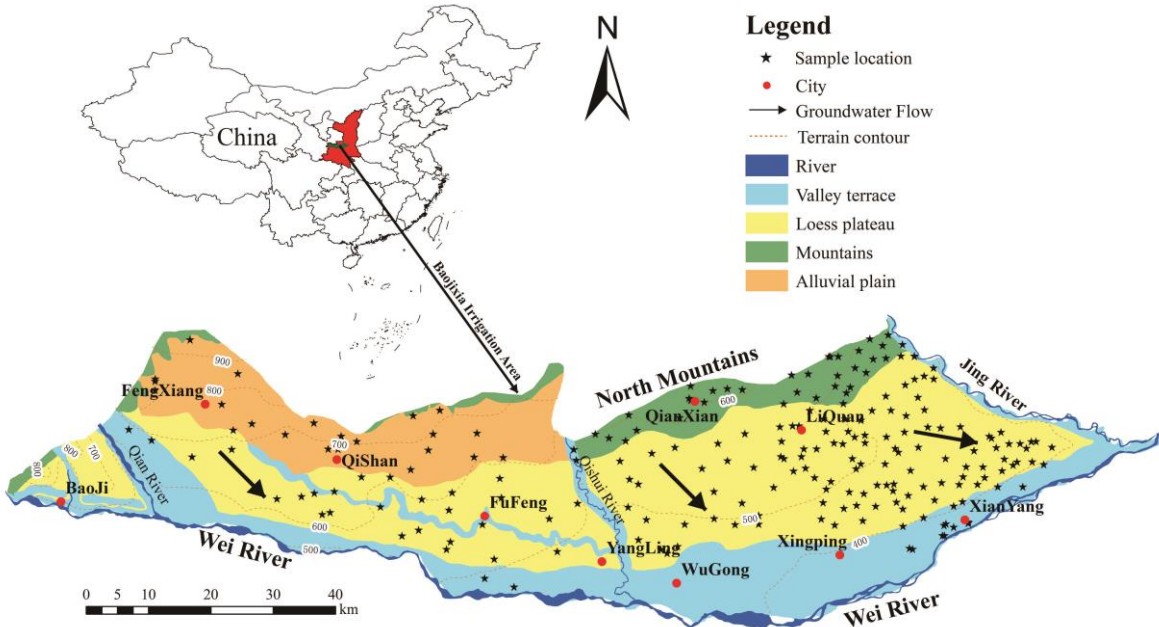

**Figure 1.** Study area and sampling locations.

The study area falls into a temperate continental monsoon climate zone and has four distinct seasons. The average annual rainfall is 600 mm, of which 50% occurs from July to September, and the minimum and maximum rainfall months are January and August, respectively [31]. The average annual potential evaporation is 1110 mm, with the minimum and maximum evaporation months being December and June, respectively. The four rivers in the study area are the Qianhe, Qishui, Jinghe, and Weihe rivers. Since the Qishui River provides a natural boundary dividing the study area into two parts, this paper uses the river as a boundary in the comparative analysis.

*2.2. Hydrogeology*

Figure 1 shows the geomorphological characteristics of the study area, which can be divided into Weihe River and the alluvial plains in its tributary, loess terraces, piedmont alluvial plains, and low mountain and hilly areas. The groundwater in the study area can be classified as mainly quaternary pore water. The lithology of the aquifer is mainly sandstone, mixed with pebbles, gravel, and clay. The aquifer mainly receives atmospheric precipitation and lateral runoff recharge in the piedmont. Within the study area, the main processes and activities reducing groundwater levels are artificial mining, discharges to the Weihe River and evaporation. There is a gradual increase in groundwater levels from the Weihe alluvial plains to the loess hills, ranging from 5–20 m and 5–80 m, respectively, with the deepest levels in the hilly area reaching 100 m [31].

## 3. Materials and Methods

### 3.1. Sample Collection and Analysis

As shown in Figure 1, a total of 223 groundwater samples were collected in the study area. Samples were collected using a standard of 125 mL polyethylene bottles as per the Chinese National Water Quality Sampling Standard [32]. Each sample bottle was washed with deionized water before sampling, and immediately before the sample collection the bottle was rinsed three times with water from the source to be sampled. The global positioning system (GPS) coordinates of the sampling points were logged during sampling, along with the depths of the well and groundwater table. The physical-chemical variables pH, electrical conductivity (EC), total dissolved solids (TDS) and water temperature (T) of the water samples were measured on-site. The samples were sent to the Water Quality Testing Laboratory of the Department of Chemical Engineering of Chang'an University for analysis. Among the variables tested for, $Na^+$ and $K^+$ were measured using flame atomic absorption spectrophotometry, $Ca^{2+}$ and $Mg^{2+}$ were measured using EDTA titration, $SO_4^{2-}$ and $Cl^-$ were determined using ion chromatography, $HCO_3^-$ was measured through acid-base titration. Finally, total hardness (TH) was calculated from the anions.

After the analysis, the charge balance error percentage (%CBE) was calculated to assess the accuracy of each sample test [33–35]:

$$\%CBE = \frac{\sum cation - \sum anion}{\sum cation + \sum anion} \times 100 \tag{1}$$

In Equation (1), all cations and anions are expressed in meq/L. It is generally accepted that a test is considered qualified if the charge balance error percentage (%CBE) < 5%. In the present study, the %CBEs of all the samples were less than 5%.

### 3.2. Irrigation Quality Assessment

The term 'soil alkalization' generally refers to the process in which sodium ions in the soil water are absorbed by the soil absorption complex. This process is usually carried out by cation exchange, i.e., the process in which sodium ions in the solution are exchanged with other cations on the surface of the soil colloid [36]. $Na^+$ in the irrigated water enters the soil solution during irrigation. A high irrigation water $Na^+$ may result in secondary alkalization of the soil, especially in the arid and semi-arid Baojixia irrigation area. In addition to total salt content, sodium content is an important irrigation water quality indicator. The sodium adsorption ratio (SAR) and the %Na- Soluble sodium percentage are often used as important indicators of the sodium content in irrigation water or soil solution. The permeability index is used to measure the long-term effect of irrigation with water containing high concentrations of $Na^+$, $Ca^{2+}$, $Mg^{2+}$ and $HCO_3^-$ on soil permeability.

Besides, certain comprehensive methods are used to evaluate irrigation water quality, such as the US Salinity Laboratory (USSL) [37] and Wilcox diagrams [38], which are simple ways of linking salinity and alkalinity to irrigation water quality assessment, and are therefore widely used in research. By comparing the sample test results with the Chinese National Standard [15] and the World Health Organization (WHO) [39] irrigation drinking water standards, the sodium adsorption ratio (SAR) and permeability index (PI) were calculated to assess the irrigation water quality in the area. The formulae of the indices are as follows [34,40]:

$$SAR = \frac{Na^+}{\sqrt{(Ca^{2+} + Mg^{2+})/2}} \tag{2}$$

$$PI = \frac{Na^+ + \sqrt{HCO_3^-}}{Ca^{2+} + Mg^{2+} + Na^+} \times 100\% \tag{3}$$

In Equations (2) and (3), all ions are expressed in meq/L.

*3.3. Relative Production Calculation*

The relative yield of soil crops with high salt content can be calculated by the following formula [1,14,22]:

$$Relative\ Yield = 100 - R(EC_s - EC_{th}) \tag{4}$$

In Equation (4), $R$ is the rate of yield loss above the threshold salinity ($EC_{th}$) and $EC_s$ is the salinity of the soil.

## 4. Results

*4.1. Groundwater Chemistry*

Table 1 is a statistical summary of the water chemical compositions of the 223 samples collected. The average pH values of the samples in the study area were 7.61 and 7.65, respectively, indicating that the groundwater is weakly alkaline. The TDS in the western region was between 203–888 mg/L, with an average value of 542 mg/L, indicating that the groundwater in the western region is fresh water. The TDS in the eastern region fluctuated greatly, with an average value of 763 mg/L, indicating that the water quality of most sampling sites was fresh water. The average value of groundwater TH in the study area is 207.0 and 23.5 mg/L, respectively, indicating that the water quality in the eastern area is relatively soft. Judging from the calculation method, hardness reflects the relationship between cation and bicarbonate in water and ion content [10,41]. In the study area, from west to east, the average $Ca^{2+}$ in cations decreased from 63.43 to 52.54 mg/L, the average $Mg^{2+}$ ion increased from 27.93 to 56.10 mg/L, and the average $Na^+$ increased from 46.41 to 220.02 mg/L. The significant increase in $Na^+$ ions may cause changes in the water chemistry types in the eastern region. The average value of $HCO_3^-$ in anions increased from 364.77 to 526.33 mg/L, the average value of $SO_4^{2-}$ increased from 16.4 to 148.99 mg/L, and $Cl^-$ increased from 26.12 to 123.60 mg/L. Abnormal concentrations of anions are usually found near clusters of human and industrial activities such as Yangling and Xianyang, indicating that human activities have affected the hydrochemical characteristics of groundwater [10,41].

**Table 1.** Physicochemical groundwater characteristics. EC: electrical conductivity; TDS: total dissolved solids; WHO: World Health Organization.

| | $Ca^{2+}$ (mg/L) | $Mg^{2+}$ (mg/L) | $Na^+$ (mg/L) | $HCO_3^-$ (mg/L) | $SO_4^{2-}$ (mg/L) | $Cl^-$ (mg/L) | TDS (mg/L) | pH | TH (mg/L) | EC (μs/cm) |
|---|---|---|---|---|---|---|---|---|---|---|
| **West of Qishui River (WQ) *n* = 51** | | | | | | | | | | |
| Minimum | 22.04 | 2.65 | 11.03 | 207.46 | 0 | 4.96 | 203 | 7.3 | 4.77 | 316.72 |
| Maximum | 195.39 | 97.86 | 133.72 | 594.9 | 127.28 | 292.49 | 888 | 8.1 | 522 | 1387.5 |
| Mean | 63.43 | 27.93 | 46.41 | 364.77 | 16.4 | 26.12 | 543 | 7.61 | 207.01 | 847.26 |
| Standard deviation | 26.81 | 14.35 | 25.68 | 62.24 | 23.31 | 44.06 | 151.26 | 0.21 | 93.48 | 236.35 |
| **East of Qishui River (EQ) *n* = 172** | | | | | | | | | | |
| Minimum | 6.01 | 2.67 | 2.07 | 225.76 | 7.20 | 8.80 | 252 | 7.10 | 5.19 | 393.75 |
| Maximum | 3701.00 | 307.55 | 700.72 | 979.32 | 1261.29 | 1680.00 | 3208 | 9.60 | 487.90 | 5012.95 |
| Mean | 52.54 | 56.10 | 220.02 | 526.33 | 148.99 | 123.60 | 763 | 7.65 | 23.53 | 1191.88 |
| Standard deviation | 280.71 | 49.86 | 98.97 | 139.71 | 171.34 | 201.17 | 466.46 | 0.33 | 43.74 | 728.84 |
| National standard | - | - | 200 | - | 250 | 250 | 1000 | 6.5–8.5 | 450 | - |
| WHO standard | - | - | 200 | - | 250 | 250 | 1000 | 6.5–8.5 | 500 | - |

*4.2. Hydrogeochemical Facies*

Piper diagrams [28] are often used to analyze the changes in and spatial distribution of ions in groundwater, as shown in Figure 2a. The distribution area of the diagrams indicates that most of the samples fall into the $HCO_3$-Na and $HCO_3$-Ca types, with a few being of the $HCO_3\cdot SO_4$-Ca·Mg mixed types. The $HCO_3$-Ca type samples were mainly concentrated in the Loess Plateau, which can be explained by the undulating topography resulting in rapid groundwater flow, thereby resulting in mineral dissolution. The $HCO_3$-Na type samples were concentrated in low-lying regions, such as river

terraces, and can be explained by evaporation and partial cation exchange. The HCO$_3$·SO$_4$-Ca·Mg and HCO$_3$·SO$_4$·Cl-Ca·Mg type samples had a patchy distribution, mostly concentrated in human settlements and industrial production areas. This suggests that fecal emissions from humans and animals may be one driver of the increases in Cl$^-$ and SO$_4{}^{2-}$ concentrations [42,43]. From west to east, the type of water chemistry gradually changed from HCO$_3$-Ca to HCO$_3$-Na and HCO$_3$·SO$_4$-Ca·Mg. This can be explained by the dissolution of sodium-containing minerals and the alternate adsorption of cations, resulting in a decrease and increase in the percentages of Ca$^{2+}$ and Na$^+$ in the groundwater, respectively. Among the anions, Cl$^-$ and SO$_4{}^{2-}$ continued to increase in varying degrees, whereas HCO$_3{}^-$ decreased slightly, showing that groundwater dissolved rock salt and gypsum during the flow process, accompanied by weak evaporation [6,10,44]. In general, the groundwater in the study area showed typical regional groundwater recharge characteristics of high HCO$_3{}^-$ and Ca$^{2+}$, low TDS and weak alkaline pH [6,34,43].

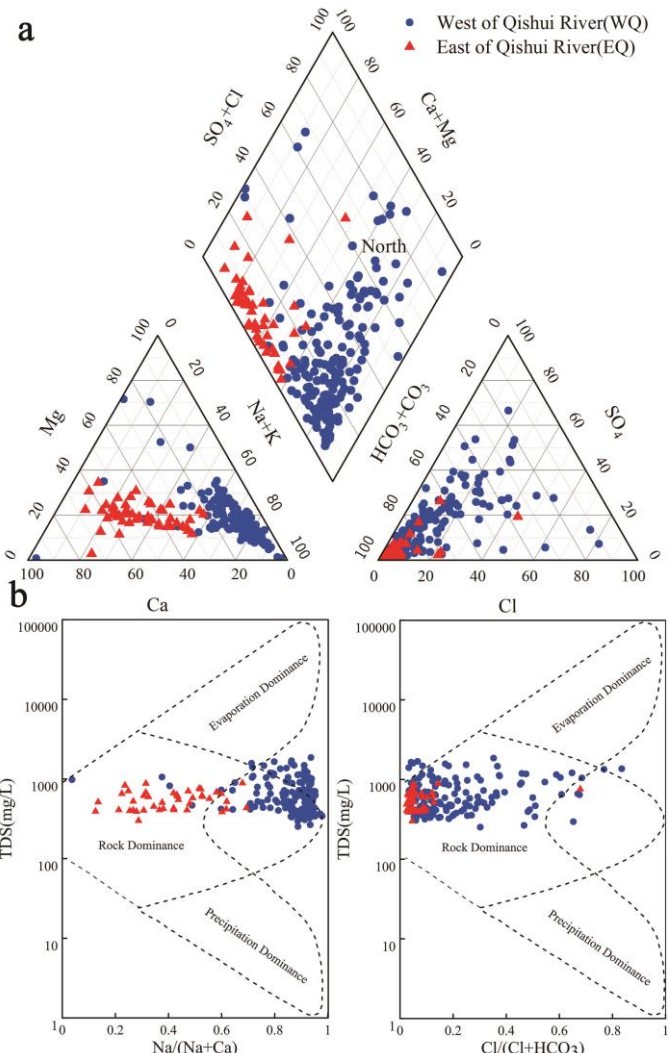

**Figure 2.** Piper (**a**) and Gibbs (**b**) diagrams representing groundwater types for different groundwater groups.

Gibbs diagrams [27] were used to represent the relationships between anions [Cl/(Cl+HCO$_3$)] and cations [Na/(Na+Ca)] and TDS in groundwater samples. These relationships can shed light on the natural origin of groundwater chemistry and the sources of dissolved chemical constituents in groundwater, such as precipitation, geology, and evaporation [15,45]. A shortcoming of this approach is that it completely ignores the impact of human activities on groundwater quality [1,2]. Figure 2b

shows that the origins of the water chemistry of all water samples on the west side of Qishui River were dominated by geology, indicating that geology is the main factor affecting the chemical characteristics of shallow groundwater in the area. Evaporation was found to be the main driver of the water chemistry of groundwater in the shallow unsaturated zone east of the Qishui River. During the evaporation of soil water into the atmosphere, the various ions contained in the water remain in the soil, resulting in increased ion concentrations in groundwater. This cumulation results in the precipitation of some less soluble minerals, and these precipitates eventually invade the saturated zone, increasing groundwater salinity. In general, the groundwater body and geology were found to be the main drivers of the hydrochemical characteristics of shallow groundwater in the study area [6,46,47].

### 4.3. Suitability for Irrigation Purpose

The applicability of groundwater in the study area was evaluated according to the irrigation drinking water standards provided by the Chinese National Ministry of Health [48] and WHO [39]. Groundwater in the Baojixia irrigation area is currently used for urban domestic uses, industry, and agricultural irrigation, with agricultural irrigation accounting for >60% of the total water consumption. The accumulation of salts and alkalinity in the soil must be considered when groundwater is used for irrigation in the study area. Indices to evaluate the potential risks shown in Table 2, including the sodium adsorption ratio (SAR) and electrical conductivity (EC) [49,50]. SAR is used to study soil sodium hazards. EC and total concentration were used to study the extent of salt accumulation in the soil. Plant growth is directly affected by the salt content of irrigation water, and indirectly by the permeability of the soil structure and aeration [6,19,49]. Therefore, PI was used to study the effect of irrigation water quality on soil permeability.

**Table 2.** Classification of groundwater samples of the study area for irrigation purposes.

| Parameters (meq/L) | Sample Range | | | | Range | Classification | Number of Samples |
|---|---|---|---|---|---|---|---|
| | Min | Max | Average | STD | | | |
| Alkalinity hazard (SAR) [37] | 0.04 | 26.79 | 4.82 | 3.35 | <10 | Excellent | 223 |
| EC | 361.72 | 2636.5 | 1017.314 | 482.26 | 10~18 | Good | 12 |
| | | | | | 18~26 | Doubtful | 0 |
| | | | | | >26 | Unsuitable | 1 |
| | | | | | <250 | Excellent | 24 |
| | | | | | 250~750 | Good | 38 |
| | | | | | 750~2250 | Acceptable | 66 |
| | | | | | >2250 | Unacceptable | 19 |

The use of a single evaluation index often results in a one-sided assessment. The USSL [37] and Wilcox diagrams [38] are used by the American Salinity Laboratory to link salinity and alkalinity to irrigation water quality assessment and are widely used in research. USSL results are shown in Figure 3a, the sample point west of Qishui River falls within the C2S1 and C3S1 categories, illustrating that the groundwater is of low alkalinity and high salinity and is suitable for irrigation. The majority of the samples to the east of the Qishui River fell within the C2S2 and C3S3 categories, indicating a medium salinity and medium-high alkalinity. The use of this quality of groundwater for an extended period can result in land salinization and is therefore not suitable for agricultural irrigation. Judging from the landform types and groundwater runoff conditions in the distribution area, the suitability of groundwater for irrigation decreases from the west to the east of the study area. The main reason for this is that as the terrain becomes lower, groundwater runoff intensity decreases, and the leaching effect weakens. On the east side of the Qishui River, which is also a slow runoff area, the rank of the suitability of different terrain for groundwater irrigation is hilly area > Loess Plateau > Weihe terrace. As the surface elevation decreases, evaporation increases. The main concentrations of human habitation and

industry occur on the loess and river terraces, resulting in higher degrees of anthropogenic influence on groundwater [10,19,43]. The Wilcox plots show that differences in the sample distributions are similar to the USSL. The plots show that groundwater quality in the western part of the study area is better suited for irrigation, whereas other regions are not suitable for long-term irrigation due to the high salinity and alkalinity of groundwater.

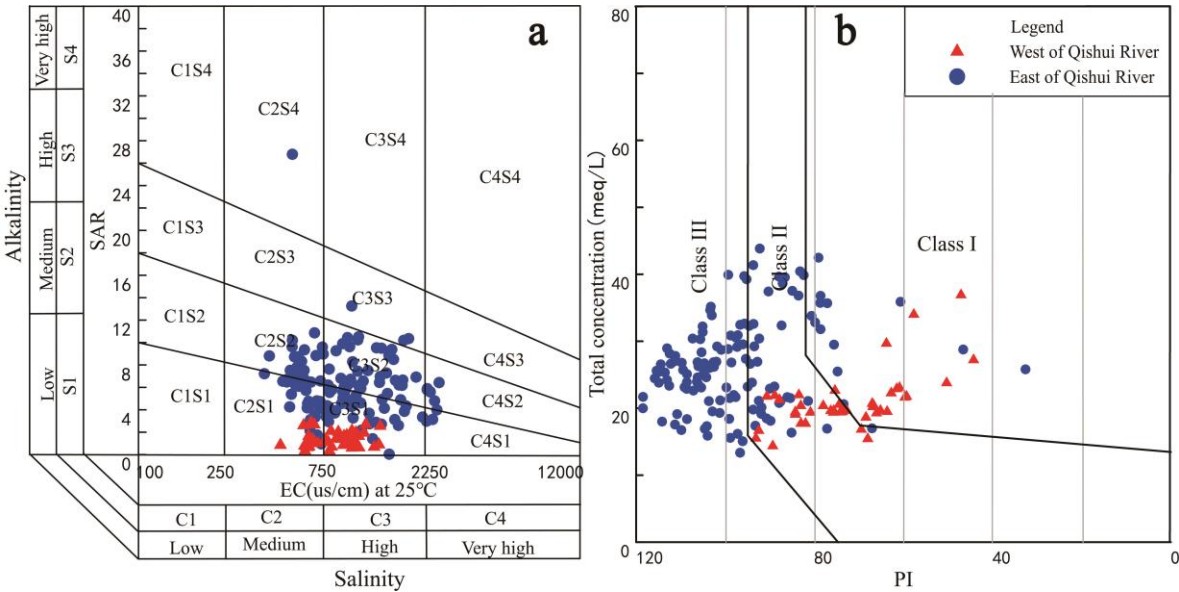

**Figure 3.** Irrigation water quality assessment based on USSL diagram (**a**) and Wilcox diagram (**b**).

Soil permeability is affected by the water contents of salts such as sodium and bicarbonate [51]. Doneen [52] developed an empirical formula (PI) to assess the impact of irrigation water on soil permeability. According to the results of the calculated PI, irrigation water quality is divided into three categories, under which categories I and II are suitable for irrigation and category III is not suitable for irrigation. As shown in Figure 3b, the vast majority of groundwater samples west of Qishui River fall within category II, with only a small proportion falling within category I. This shows that groundwater in the area is suitable for irrigation. Of the samples collected in Liquan and Qianxian, 93.4% fell within categories I and II, indicating that groundwater in this area is suitable for irrigation. The remaining 6.6% in category III was mostly concentrated in areas of human settlement and industrial activity, and these areas are not suitable for irrigation. Of the groundwater samples collected in Xianyang and Xingping, 25.4% fell within categories I and II. Most of these samples were located in areas where the hydrological cycle alternates rapidly, such as rural areas and piedmont alluvial fans. Groundwater in these areas is suitable for irrigation. The remaining 74.6% of the water samples falling within category III indicate areas not suitable for irrigation. This may be because Xianyang and Xingping are located on the Weihe terrace where groundwater is recharged by the polluted Weihe water.

## 5. Discussion

### 5.1. Hydrogeochemical Characteristics

Statistical analysis usually yields better conclusions than graphical methods when studying the dominant factors influencing groundwater chemistry. This is because the statistical analysis can identify the relationships between the variables to reflect actual problems. Correlation matrix and factor analysis are two such statistical methods commonly used in groundwater hydrogeochemical research [6,7].

### 5.1.1. Correlation Matrix

The correlation matrix of groundwater ions is shown in Table 3. Strong correlations were evident between $Cl^-$ and $Mg^{2+}$ (0.771) and between $Cl^-$ and $Na^+$ (0.611) in groundwater. Strong correlations were also evident between $SO_4^{2-}$ and $Mg^{2+}$ (0.631) and $SO_4^{2-}$ and $Na^+$ (0.662) ions. This indicates that water-rock interaction and mineral dissolution may be the main hydrogeochemical processes driving groundwater chemistry in the study area. A strong negative correlation was evident between $Na^+$ and $Ca^{2+}$ (−0.400) ions, indicating opposite ionic trends between the two in groundwater, which may be due to cation exchange between $Na^+$ and $Ca^{2+}$. A strong positive correlation was evident between $Na^+$ and $Mg^{2+}$ (0.431), indicating that $Na^+$ and $Mg^{2+}$ may both originate from mineral dissolution. These results show that the main exchange between $Ca^{2+}$ in groundwater in the study area is with $Na^+$, which is consistent with the conclusions drawn from the previous analysis of hydrochemical types. A strong positive correlation was evident between $Cl^-$ and $SO_4^{2-}$ (0.667) ions, and both had strong correlations with $Na^+$ (0.611 and 0.662, respectively) and $Mg^{2+}$ (0.771 and 0.631, respectively).

**Table 3.** Matrix plot of the physicochemical groundwater characteristics.

| Hydrochemical Parameter | pH | EC | $Cl^-$ | TDS | $Ca^{2+}$ | $Mg^{2+}$ | $Na^+$ | $HCO_3^-$ | $SO_4^{2-}$ |
|---|---|---|---|---|---|---|---|---|---|
| pH | 1.0000 | 0.2517 | 0.1401 | 0.2517 | 0.0736 | 0.0481 | −0.0754 | −0.3106 | −0.0394 |
| EC | 0.2517 | 1.0000 | 0.1988 | 0.8634 | −0.0223 | 0.1317 | 0.0530 | −0.1708 | 0.1365 |
| $Cl^-$ | 0.1401 | 0.1988 | 1.0000 | 0.1988 | 0.1632 | 0.7706 | 0.6109 | −0.0911 | 0.6670 |
| TDS | 0.2517 | 0.8634 | 0.1988 | 1.0000 | −0.0223 | 0.1317 | 0.0530 | −0.1708 | 0.1365 |
| $Ca^{2+}$ | 0.0736 | −0.0223 | 0.1632 | −0.0223 | 1.0000 | 0.2719 | −0.4003 | −0.4584 | 0.0405 |
| $Mg^{2+}$ | 0.0481 | 0.1317 | 0.7706 | 0.1317 | 0.2719 | 1.0000 | 0.4312 | 0.1123 | 0.6311 |
| $Na^+$ | −0.0754 | 0.0530 | 0.6109 | 0.0530 | −0.4003 | 0.4312 | 1.0000 | 0.4273 | 0.6618 |
| $HCO_3^-$ | −0.3106 | −0.1708 | −0.0911 | −0.1708 | −0.4584 | 0.1123 | 0.4273 | 1.0000 | 0.0113 |
| $SO_4^{2-}$ | −0.0394 | 0.1365 | 0.6670 | 0.1365 | 0.0405 | 0.6311 | 0.6618 | 0.0113 | 1.0000 |

### 5.1.2. Factor Analysis

Factor analysis can explain the molar relationship between various factors controlling hydrogeochemical reactions in groundwater [14,29]. This method is one of the most useful for understanding the hydrogeochemical evolution of groundwater. In statistical analysis, a large proportion of chemical data is transformed into a small number of factors. During this process, the load of each variable is converted into a factor, and these factors are then extracted using principal component analysis [14,53,54]. The few factors with the most relevant principal components are used to represent the diversity of geochemical data and structures. Variance and covariance among geochemical elements are used to correlate factors and derive their relationships. Table 4 shows that the factors influencing groundwater chemistry in the study area can be divided into three main components, whose cumulative contribution is 75.63%. These three principal components can be used to represent the main factors influencing groundwater quality in the study area. Table 5 shows that the most relevant ions in the first principal component are $Cl^-$, $Mg^{2+}$, $Na^+$, and $SO_4^{2-}$, whereas those in the second principal component are $Ca^{2+}$ and $HCO_3^-$, whereas pH and TDS are those in the third principal component. These results show that the main cations influencing groundwater chemical composition in the study area are $Na^+$, $Ca^{2+}$, and $Mg^{2+}$ and the anions are $Cl^-$, $SO_4^{2-}$, and $HCO_3^-$.

**Table 4.** Factor total variance interpretation.

| Factors | Initial Eigenvalue | | | Extract Load Sum of Squares | | | Sum of Rotation Load Squares | | |
|---------|-------|------------------------|-------------------|-------|------------------------|-------------------|-------|------------------------|-------------------|
| | Total | Percentage Variance | Accumulate (%) | Total | Percentage Variance | Accumulate (%) | Total | Percentage Variance | Accumulate (%) |
| 1 | 2.95 | 36.85 | 36.85 | 2.95 | 36.85 | 36.85 | 2.89 | 36.11 | 36.11 |
| 2 | 1.92 | 23.98 | 60.83 | 1.92 | 23.98 | 60.83 | 1.79 | 22.33 | 58.44 |
| 3 | 1.18 | 14.80 | 75.63 | 1.18 | 14.80 | 75.63 | 1.38 | 17.19 | 75.63 |
| 4 | 0.73 | 9.17 | 84.80 | | | | | | |
| 5 | 0.60 | 7.49 | 92.29 | | | | | | |
| 6 | 0.31 | 3.82 | 96.12 | | | | | | |
| 7 | 0.23 | 2.93 | 99.04 | | | | | | |
| 8 | 0.08 | 0.96 | 100.00 | | | | | | |

**Table 5.** Correspondence between principal components and indicators.

| Factors | 1 | 2 | 3 |
|---|---|---|---|
| pH | 0.040 | 0.513 | 0.590 |
| TDS | 0.221 | 0.332 | 0.671 |
| $Ca^{2+}$ | 0.008 | 0.747 | −0.546 |
| $Mg^{2+}$ | 0.835 | 0.196 | −0.230 |
| $Na^+$ | 0.796 | −0.463 | 0.160 |
| $Cl^-$ | 0.894 | 0.239 | −0.033 |
| $HCO_3^-$ | 0.158 | −0.823 | −0.014 |
| $SO_4^{2-}$ | 0.862 | 0.025 | −0.083 |

*5.2. Major Factors Influencing Groundwater Chemistry*

5.2.1. Water-Rock Interaction

Water-rock interaction is one of the main processes driving the concentration of chemical constituents in groundwater. The relative proportions of various dissolved ion concentrations in groundwater depend to a large extent on the solubility of rocks in aquifers [2,34,46]. Figure 4 shows graphs representing the relationships between the concentrations of pairs of geochemical ions, which can assist in analyzing and understanding the water-rock interaction process that controls groundwater chemistry. If both $Na^+$ and $Cl^-$ in the solution originate from the dissolution of salt (NaCl), the $Na^+$ and $Cl^-$ concentration values in a groundwater sample should be on the 1: 1 line (R1). However, almost all the water sample points shown in Figure 4a are above the 1:1 line, suggesting that most $Cl^-$ in this area may originate from the dissolution of salt. The growth rate of $Na^+$ along the groundwater flow direction was significantly higher than that of $Cl^-$, indicating that $Na^+$ may originate from other sources, such as cation exchange or silicate weathering (R2) [6,55]. Figure 4b shows that the distributions of $SO_4^{2-}$ and $Ca^{2+}$ in the study area gradually shifted below the 1:1 line, with $SO_4^{2-}$ increasing faster than $Ca^{2+}$. This suggests that in addition to gypsum dissolution (R3), other factors such as cation exchange result in the loss of $Ca^{2+}$. Besides, if human activities result in the input of $SO_4^{2-}$ into groundwater at a faster rate than that by the dissolution of gypsum and anhydrite, $SO_4^{2-}$ will increase at a rate faster than that of $Ca^{2+}$ to a certain extent [10,34,56].

The dissolution of calcite and dolomite can add $Ca^{2+}$ and $HCO_3^-$ to groundwater (R4 and R5). Under a situation of all $Ca^{2+}$ originating from the dissolution of calcite or dolomite, samples would fall near the 1:1 line or 2:1 line. Figure 4c shows that some samples plotted along the expected zone whereas some fell below it. Meanwhile, $Ca^{2+}$ concentration decreased as $HCO_3^-$ concentration increased. This indicates that while the dissolution of calcite and dolomite is indeed the source of $Ca^{2+}$, other factors also affect the $Ca^{2+}$ concentration in groundwater [29,49]. Figure 4d shows that as $HCO_3^-$ concentration increases, the positions of sample change from near the 1:1 line to below the 2:1 line. The difference between Figure 4d,c relates to the $Mg^{2+}$ concentration value. In a situation where all dissolved $Mg^{2+}$ in the groundwater originates from dolomite, an equal amount of $Ca^{2+}$ would enter the water (R4) [14,41]. If it is assumed that all $Ca^{2+}$ in Figure 4c originates from calcite, the points in Figure 4d should be above the 1:1 line (R5), whereas they fall below the 1:1 line, indicating that the dissolution of dolomite in groundwater produced $Ca^{2+}$, $HCO_3^-$ and $Mg^{2+}$, while at the same time inhibiting the dissolution of partial calcite as a reverse reactant (R5), thereby reducing the concentrations of $Ca^{2+}$ and $HCO_3^-$ in groundwater. Besides, the concentrations of $Ca^{2+}$ and $HCO_3^-$ decrease, with dolomite continuing to dissolve before reaching the dissolution equilibrium, which increases the $Mg^{2+}$ content in the groundwater (R4) [42,46,57].

$$NaCl \rightarrow Na^+ + Cl^- \tag{R1}$$

$$NaAlSi_3O_8 + 8H_2O \rightarrow Na^+ + Al(OH)_4^- + 3H_3SiO_4 \tag{R2}$$

$$CaSO_4 + 2H_2O \rightarrow Ca^{2+} + SO_4^{2-} + 2H_2O \tag{R3}$$

$$CaMg(CO_3)_2 + CO_2 + H_2O \rightarrow Ca^{2+} + Mg^{2+} + 4HCO_3^- \tag{R4}$$

$$CaCO_3 + CO_2 + H_2O \rightarrow Ca^{2+} + 2HCO_3^- \tag{R5}$$

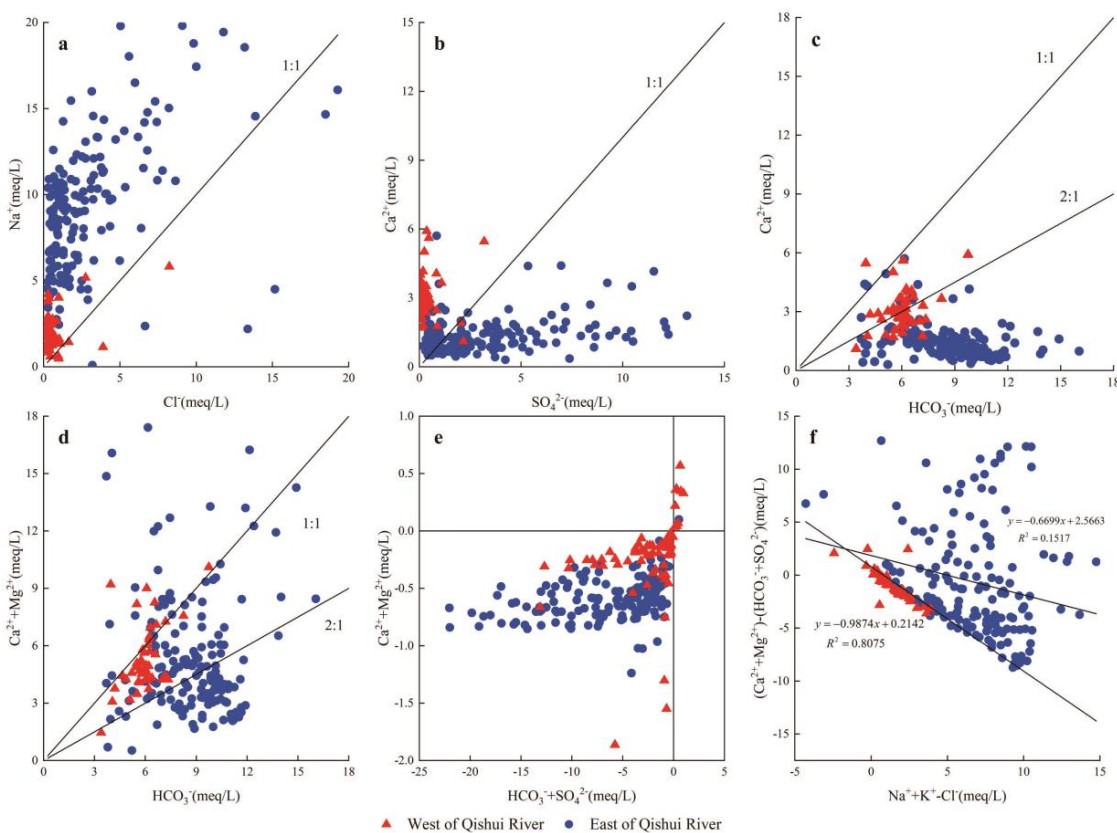

**Figure 4.** Bivariate diagrams of major ionic concentrations in groundwater samples (**a**): Na$^+$ vs. Cl$^-$; (**b**): Ca$^{2+}$ vs. SO$_4^{2-}$; (**c**): Ca$^{2+}$ vs. HCO$_3^-$; (**d**): Ca$^{2+}$ + Mg$^{2+}$ vs. HCO$_3^-$; (**e**): Ca$^{2+}$ + Mg$^{2+}$ vs. HCO$_3^-$ + SO$_4^{2-}$; (**f**): (Ca$^{2+}$ + Mg$^{2+}$) − (HCO$_3^-$ + SO$_4^{2-}$) vs. Na$^+$ + K$^+$ − Cl$^-$.

### 5.2.2. Cation Exchange

Cation exchange represents an additional chemical mechanism occurring in the aquifer that affects the ion concentrations of groundwater. This process acts as a temporary buffer when the ion concentration changes [6,49]. The main ions involved in ion exchange are Na$^+$ (or K$^+$) and Ca$^{2+}$ (or Mg$^{2+}$) (Equations (R6)–(R9)), which results in opposite trends in the concentrations of Na$^+$ and Ca$^{2+}$, and has a great impact on the chemical characteristics of groundwater.

$$Ca^{2+} + 2NaX \rightarrow 2Na^+ + CaX_2 \tag{R6}$$

$$Mg^{2+} + 2NaX \rightarrow 2Na^+ + MgX_2 \tag{R7}$$

$$Ca^{2+} + 2KX \rightarrow 2K^+ + CaX_2 \tag{R8}$$

$$Mg^{2+} + 2KX \rightarrow 2K^+ + MgX_2 \tag{R9}$$

Schoeller [58] proposed two chlor-alkali indices, CAI-1 and CAI-2, for studying cation exchange, defined as:

$$CAI - 1 = \frac{Cl^- - (Na^+ + K^+)}{Cl^-} \tag{5}$$

$$CAI - 2 = \frac{Cl^- - (Na^+ + K^+)}{HCO_3^- + SO_4^{2-} + CO_3^- + NO_3^-} \tag{6}$$

When $Na^+$ and $K^+$ adsorbed on the surface of the aquifer particles are replaced by $Ca^{2+}$ and $Mg^{2+}$ in the flowing groundwater, both CAI-1 and CAI-2 are less than 0 [6,49]. When reverse ion exchange occurs, CAI-1 and CAI-2 are both greater than 0. As shown in Figure 4e, 95% of samples showed CAI-1 and CAI-2 less than 0. This result indicates that $Na^+$ and $K^+$ ions in the aquifer are replaced by $Ca^{2+}$ and $Mg^{2+}$ in the groundwater, and the dominant cation changed from $Ca^{2+}$ to $Na^+$. Figure 4f is a bivariate diagram of $(Na^+ + K^+ - Cl^-)$ and $(Ca^{2+} + Mg^{2+}) - (SO_4^{2-} + HCO_3)$, which can also be used to characterize the cation exchange process in groundwater. Here, $(Na^+ + K^+ - Cl^-)$ represents the amount of $Na^+$ added or lost by the dissolution of gypsum, anhydrite, calcite, and dolomite. When the cation exchange process is the main process controlling the chemical characteristics of groundwater, the two parameters are inversely proportional and the slope is close to −1. Figure 4f shows that the slope of the curve for the samples on the west side of Qishui River is −0.9874, whereas that for those collected on the east side is −0.6699, both of which are approximately equal to −1. This indicates that significant cation exchange occurred in the study area in groundwater. Cation exchange maybe is a dominant driver of changes in groundwater chemical type from the $HCO_3$-Ca type to $HCO_3$-Na type.

### 5.3. Water Quality Influence on Crop Yields

Salt from soil and groundwater are absorbed by root tissues during plant production. High concentrations of $Na^+$ and $HCO_3^-$ expose plants to salt and alkali hazards. Some plants suffer foliar damage due to exposure to water containing high concentrations of $Na^+$ and $Cl^-$, which has an impact on plant growth [8,15,18,59]. The exchangeable sodium ratio (ESP) was used to measure the tolerance of a crop to salt. The salinity tolerances of different crops such as corn and wheat are typically low during the early stages of seed germination, but increase in later stages [14,22]. Since the dominant crops of the study area are mainly wheat and corn, Table 6 shows the yield potential (%) of these two crops after irrigation, according to Ayers' study [22]. The relative crop yields of wheat and corn were calculated based on limits to crop salt-tolerances (100%) and groundwater quality of the study area, using the relative yield formula. The relative crop yields were converted to the annual relative yield loss rates (RAYLR) of wheat and corn, as shown in Figure 5a, b, respectively. The results showed that the area of relative yield loss rates (RAYLR) < 5% accounted for 53.65% and 28.33% of annual relative yield loss rates of wheat and corn, respectively. Meanwhile the area of relative yield loss rates (RAYLR) between 5% and 10% accounted for 26.18% and 27.90%, respectively, and the area of RAYLR > 10% accounted for 20.17%, and 43.78%, respectively.

**Table 6.** Yield potential of Wheat and Corn [22].

| Crops Botanical Name | Common Names | Yield Potential (Salinity Tolerance Limit) | | | |
|---|---|---|---|---|---|
| | | 100% | 90% | 75% | 50% |
| Zea mays | Corn | 75.17 (1.1) | 90.60 (1.7) | 95.30 (2.5) | 97.99 (3.9) |
| Triticum | Wheat | 99.30 (4.0) | 100 (4.9) | - | - |

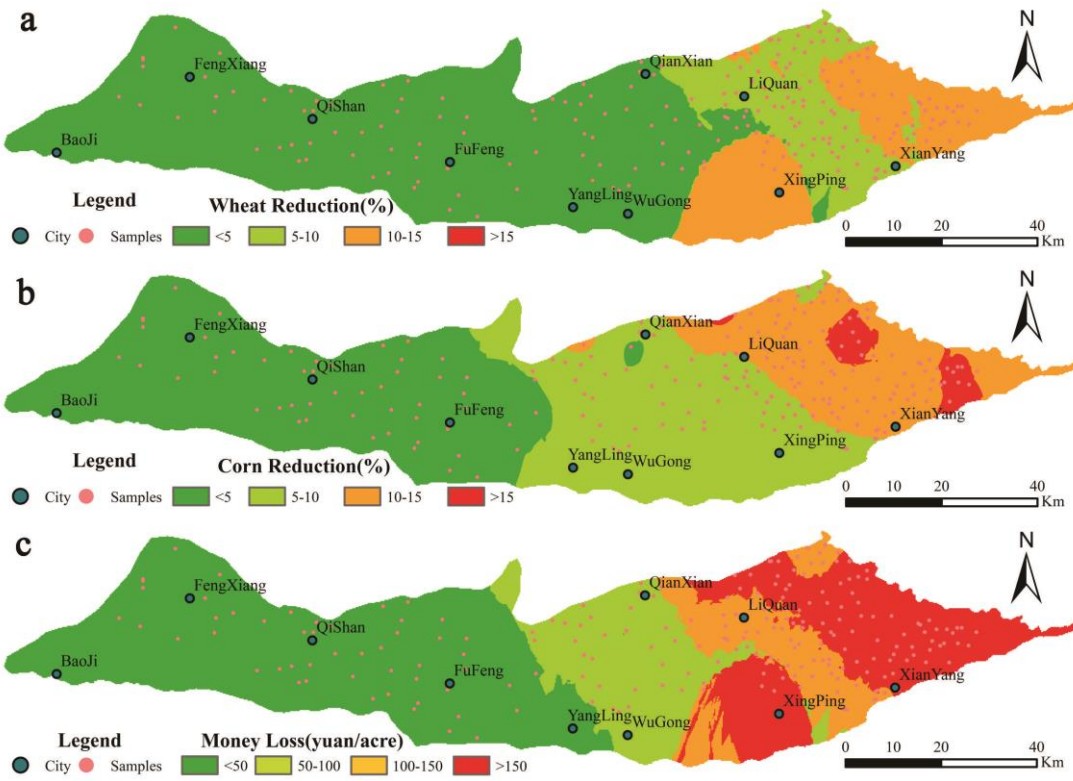

**Figure 5.** Spatial Variation map of Wheat Reduction (**a**), Corn Reduction (**b**) and Money Loss (**c**).

Figure 5c illustrates the spatial distribution of economic loss through reduced agricultural production, based on the average acre yield of crops and sales price in Xi'an in 2018. Areas with economic losses < 50 yuan accounted for 32.62% of total area, economic losses between 50 and 100 yuan accounted for 17.17% of total area, economic losses between 100 and 150 yuan accounted for 18.45% of total area and economic losses > 150 yuan accounted for 31.76% of total area. According to the 2018 Statistical Yearbook of Shaanxi Province, the total crop output in the study area was 3.081 million tons. It is estimated that the relative yield loss of the study area in 2019 could reach 0.489 tons, accounting for 0.0112% of crop yield in 2018. The economic loss was $0.740 \times 10^4$ yuan in 2019, accounting for 0.1698% of the GDP of the study area in 2018.

Since the improvement of groundwater quality takes a long time, the use of inferior groundwater will have to continue in the short term. The formula for calculating the relative loss of continuous production is $0.489 \times (1 + 0.0112\%)^{N \text{ (years)}}$ tons, whereas the formula for calculating the economic loss is $0.741 \times (1 + 0.1698\%)^{N \text{ (years)}} \times 10^4$ yuan. As shown in Figure 6, by 2119, the cumulative loss of crop output was 49.17 tons and the cumulative economic loss was as high as $80.781 \times 10^4$ yuan. This loss accounted for 6.16% of the total food production and 0.185% of the GDP of the study area in 2018. Baojixia is the main irrigated agricultural production region in Shaanxi, and a large amount of groundwater is continuously used for irrigation every year. At the same time, human waste emissions and continuous nitrate pollution through irrigation results in further deterioration of groundwater [10,24,56]. Under these dual effects, the groundwater level constantly rises and soil salinity hazards continue to increase. As a result, crop yield losses and economic losses will further increase. With a growing population in the Guanzhong region, food security is becoming ever more important. Improving groundwater quality to ensure food security is therefore important.

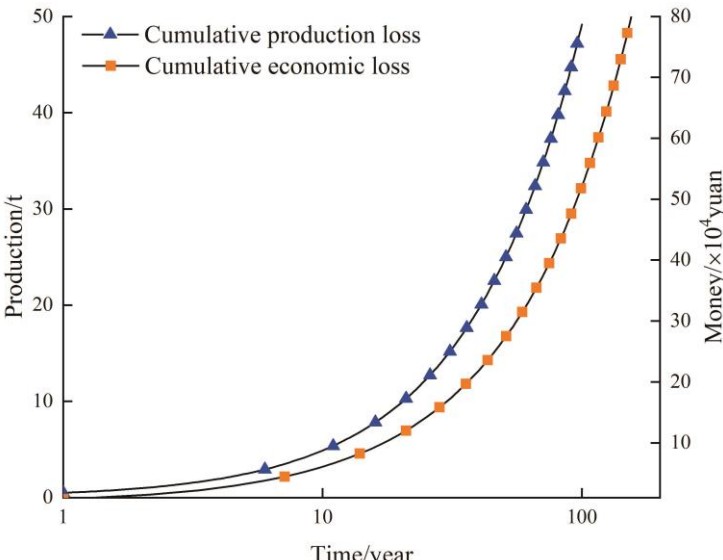

**Figure 6.** Cumulative loss of output and money over time.

## 6. Conclusions

Graphical methods, multivariate statistics and geochemical analysis were used to evaluate groundwater quality and determine its suitability for crop irrigation. Water-rock interaction and leaching were identified as the main hydrogeochemical processes affecting groundwater quality, with cation exchange being the dominant process in the eastern region. Of all the samples, 5.58% and 63.67% of SAR and EC were not good, respectively, illustrating that high salt and low sodium in shallow groundwater pose challenges to groundwater irrigation in the study area. The combined use of USSL and Wilcox charts showed that groundwater on the west side of the study area has low salinity and alkalinity, and therefore, is suitable for irrigation. In contrast, groundwater on the east side of the study area has a medium to high salinity and alkalinity and is therefore not recommended for use in long-term irrigation.

Hydrogeochemical characteristics showed that there were some significant spatial variations in some geochemical variables. The Piper diagram showed that cations in groundwater are dominated by $Na^+$, $Ca^{2+}$ and $Mg^{2+}$, whereas anions are mainly $HCO_3^-$, $SO_4^{2-}$ and $Cl^-$. From west to east, the predominant water chemistry changed from $HCO_3$-Ca to $HCO_3$-Na. Gibbs diagrams showed that water-rock interaction is the main process affecting shallow groundwater chemistry in the area. Correlation matrices and principal component analyses confirmed that water-rock interaction is the main driver of cation and anion enrichment in groundwater. At the same time, the matrices showed that cation exchange is the main reason for the replacement of $Na^+$ by $Ca^{2+}$ in groundwater. Therefore, the Chlor-alkali index was used to study the cation exchange process. Of the sampling points, 94.3% confirmed the replacement of $Ca^{2+}$ and $Mg^{2+}$ by $Na^+$ and $K^+$ in the aquifer. The slope of the fitting curve on the east side of the study area was approximately equal to −1, confirming once again that cation exchange is the main driver of groundwater chemical type. In general, the hydrogeochemistry of the study area mainly depends on the geochemical processes (weathering of rocks, cation exchange, adsorption and analysis, precipitation and evaporation) that occur in the aquifer system.

The relative yield formula determined by the Mass-Hoffman model was used to calculate the relative yield loss. Using 2018 as an example, under a scenario of irrigation of wheat and corn with groundwater in the study area, there would be a reduction in annual crop output by 0.489 tons with an economic loss of $0.741 \times 10^4$ yuan. The recovery of shallow groundwater in the Guanzhong area will take a long time. During this period, industrial and human waste and agricultural fertilizers will continue to contaminate groundwater, leading to further deterioration of water quality. Even under a scenario of groundwater quality remaining unchanged for a long time, the cumulative loss of

total crop output will reach 49.17 tons by 2119, with a cumulative economic loss of $80.781 \times 10^4$ yuan. With the intensification of climate change and human activities, surface water resources have become increasingly scarce. Therefore, large-scale exploitation and utilization of groundwater for domestic use and irrigation have become increasingly prevalent. Deteriorating groundwater quality will result in a small or insignificant loss in crop yields in the short term, but this loss will be considered in the long term.

**Author Contributions:** Conceptualization, H.Q.; Data curation, W.F.; Investigation, P.X. and K.H.; Methodology, W.F.; Project administration, H.Q.; Software, W.F. and P.X.; Writing—original draft, K.H. All authors have read and agreed to the published version of the manuscript.

**Funding:** We are grateful for the support from the National Natural Science Foundation of China (41572236).

**Acknowledgments:** This study was financially supported by the National Natural Science Foundation of China (41572236 to Hui Qian). And the completion of this article was inseparable from the contributions of all authors. Their support is gratefully acknowledged.

**Conflicts of Interest:** The authors declare no conflict of interest.

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
