# Peer review of "Hydrochemical Characteristic of Groundwater and Its Impact on Crop Yields in the Baojixia Irrigation Area, China"

_water, doi:10.3390/w12051443_

Round 1

Reviewer 1 Report

The paper entitled "Hydrochemical evolution of groundwater and its impact on crop yields in the Baojixia irrigation area, China" was submitted for review for possible publication.  So here are my comments:

1.  Since the title mentioned "evolution", it is expected that the authors present progressive analysis if the hydrochemical facies of the ground water.  If they cannot, I suggest this word must be removed and instead use the term "characteristic".

2.  Why is K+ not included in the physico chemical characteristics of the groundwater?  This was questioned because in discussion, you have conculded that illite and montmorillonite are the main components of minerals in the dissolution. Several literatures have indicated that there is very small illite occurence in China region.  On the other hand, montmorillonite is abundant in China.  So where did you get the idea that there is illite in your ground water? Please present proof. 

3.  In addition, you are hereby asked to provide substantial proof on your claims written in line 282 to 294.  The correlation between cations and anions are very week reasons for this. 

4. One of the most important data in your correlation table is the perfect positive (1.000) correlation between TDS and EC.  Explain this in the contet of the type of ground water you have in the study area,if it is good for irrigation, what are its attributes why it was good for irrigation.  

5.  Explain very well the relationship between EC and SAR.  Since the area of study: west side good for irrigation while east side, not good for irrigation, discuss the why is that so in the context of EC and SAR.  What are the implications? 

6. Based on the TDS, what type of water was your ground water will be, fresh?, brackush? saline? or brine? This is because you have presented TDs in most all of your figures so better utilize it.

7. Why used Factor Analysis instead of Principal component analysis? Factor analysis was usuallu used in social and behavioral sciences. 

8.  Based on the characteristics of your ground water. what is the order of abundance of you cation? anion? what is its implication to the groundwater?

9. In line 404 and 405 formulas, cite where you get this formulas and what is the justification why you used this formula over other formulas available.  

10.  Your figure 6 was captioned "cumulative economy loss of crop yields and economy over time", then you presented a double-y graph. Fix this one.

Author Response

Thank you for your valuable comments, please refer to the attachment for details!

Reviewer 2 Report

Comments attached

Author Response

(The authors gave the same response as above.)

Reviewer 3 Report

The manuscript reports a scientifically sound and well described analysis of ground water in the region of Baojixia. The results have been related to the crop yield in 2019 and this dependence have been forecasted during the next 100 years. The results are novel and their importance is related to the important role played by this region in the chinese crop production. The analysis over the next 100 year is only futuristic as after few years the economic losts will not justify agriculture for 100 years! This should be probably mentioned throughoyt the text. English must be improved and some misprints corrected (for instance at line 9 in the abstract " be low" must be written instead of  "below" and so on).

Once these minor points will  be corrected the paper can be published in my opinion.

Author Response

Modified, thanks for your comments!

Round 2

Reviewer 1 Report

On the comment and response on:

Point 4: One of the most important data in your correlation table is the perfect positive (1.000)
correlation between TDS and EC. Explain this in the contet of the type of ground water you have in
the study area,if it is good for irrigation, what are its attributes why it was good for irrigation.

Response 4: I am also very strange about this, but this is not the main purpose of factor analysis and
can only be left for later.For irrigation water quality, EC reflects the ion content in the water.
Generally speaking, high ion content is beneficial for plants to maintain the necessary growth ions,
which can be used as a judgment index. See Table 2 for details.

Points to consider:

TDS and EC are related to each other.  Both are indicators of salinity.  Though you have mentioned that you classify your ground water as fresh water, how will you explain this phenomena? 

You have mentioned that C2S1 - C3S1 have high salinity; low alkalinity while C2S2 - C3S3 has medium salinity and high alkalinity.  Based on this analysis, salination of freshwater is in progress.  Your ground water is changing from a typical freshwater into salinated freshwater.  If not addressed, this phenomena will definitely alter the gW composition and its classification.

Read and consider the papers below to be enlighted on EC and salinity.

 https://www.tandfonline.com/doi/abs/10.1080/10934529.2014.859035

Please fix the following errors:

In all the tables, write correctly the chemical formula for the anions and cations.  

1. In table 1, you have given anions charges while no charges indicated in the cations.

2. In table 1 fix the unit for EC, it should be microsiemens per cm not us/cm.

3. Same comments on writing correctly cations and anions formula for Table 4. Why do you have 2 Table 4 in page 11? Fix this.

5. For Table 4: replace the word "ingredients" with a more appropriate term (may be "factors" is better). You are not cooking here. 

6. For Table 4: change PH to pH.

7. Table 4: fix chemical formulas for cations and anions

8. Table 4: Check column labels "Percentage where e is dropped to the 2nd line" its an eyesore. fix it.

9. For the Conclusion:

Balance by adding and expounding the impact of the characteristics of your GW to the crop yield.  Most of it was about the characteristics of the ground water but only the 2 last lines were about the impacts to crop yield.  Discuss the impact of:

  1. highly correlated cations and anions  to the crop yield
  2. TDS, EC and the changing salinity of the groundwater to crop yield.
  3. variablity on ins concentration, will there more expected loses in the west side than the east side then how come.

Author Response

I'm very sorry for taking up your time many times. Your comments are very pertinent and valuable. I would like to express my deep gratitude! The specific reply is attached.

Reviewer 2 Report

***I am sorry, but after having a look to the responses to my comments, I have understood that I didn't attach the right file in my first revision. In fact, it was a very preliminary draft. This is why the responses don't address clearly most of my suggestions/questions, and then I am not able to make a final decision on this stage. I regret the inconveniences.

The manuscript  is a case study of a relatively large irrigated area in China, based on the major-ion composition of groundwater with the objective of quantitatively estimate loses in cereal production derived of the quality of the irrigation water. Methods used can be described as “classical”, although there are some inconsistencies and drawbacks, as well as some aspects/values that raise doubts (see attached comments).  The Interest of the results/discussion seems to me of local scope. Text is, in general, easy to follow.  The reference list seems pertinent. Illustrations are of good quality, although some of them can be simplified and improved. Abstract is OK, although it can be shortened. Conclusions should be significantly reduced.

24           “… output and an economic loss of 0.489 tons and 0.741 million yuan, respectively” (for 2019). (less than 0,5 t seems a very low value: check the figures ¿or maybe are they expressed as per surface unit?).

25           “was calculated to result in the cumulative loss of crop yields and an economic loss of 49.17 tons and 80.781 million yuan, respectively by 2119.” (this seems a repetition that can be omitted: it more or less corresponds to multiplying the previous values by 100, the time period from now. Again, the same doubt: are per surface values?)

48          “The model sets a threshold salt limit parameter to describe the salt tolerance potential of crops.” It would be good to provide, as an example, some values of this parameter for the main crops of the study area.

50           …”as the slope of the model curve increases” (this is not understandable here: clarify or delete it)

77           “geochemical simulations have been used to evaluate groundwater quality and to identify the evolution of hydrogeochemical processes.” (see comment later on)

93           “In the study area, arable land comprises ~19.54 hm2, with ~18.90 hm2 irrigated” (it has been said that the study area has an extension of some 1,800 km2. I don’t understand why these values are so reduced)

94           “The coefficient of the degree of agricultural water shortage is ~1.19” (give a brief explanation of what this coefficient is). The (L 99) it is said that “the degree coefficient of the annual average water shortage is ~39.9%” (it is unclear to what these markedly different percentages refer)

95           “Approximately 63.19% and 55.24% of total surface water and groundwater are used in the irrigation area, respectively” (explain the absolute values to which these percentages apply)

99           ~143 million m3, ~463 million and ~119 million m3 “ (why not hm3? to be consistent with the previous use of hm2)

Map (Fig. 1). Does the map correspond to the Baojixia irrigation area, or to the Guanzhong Plain? (if not, situate conveniently these two areas, which are mentioned in the introduction ad later on).  It must be supposed (if the reader is enough good in geography) that the contours of the big map in black and white are those of China, and that the internal divisions correspond to the provinces, and that the blue one is the Shaanxi Province (am I right?). I recommend not presenting a so “mute” location map. Furthermore, I have identified in the map 6 rivers labelled: Jing, Gan, Qishui, Qian, Jingling and Wei. Before, it has been said that the main rivers were the Qianhe, Qishui, Jinghe and Weihe . I notice only one coincidence (this should be clarified). Moreover, the blue colour of these labels makes difficult the reading.

The geographical indication “Loess Plateau” is indicated in other parts of the text. I suggest labelling this territory in the map.

“Calcium” (¿calcrete?) and “sediment” in the legend seem vague or inappropriate terms. The N and Qs signs in the cross section are not explained in the legend. Concerning the “rich”, “good” and other similar terms in the right extreme of the legend: I suppose that they refer to the aquifer productivity; if so, explain (but I think that this information is not used in the context of this manuscript, so I suggest delete this colour code). In general, I don`t see the usefulness, in the context of the manuscript, of differentiating in the cross-section materials such as pebbles, gravels, coarse sands, etc. I suggest making it as simpler as possible concerning lithostratigraphy.

Unfortunately the map doesn`t include some isopotentiometric curves that can help to understand more precisely the groundwater flow paths. Concerning this, there are only three arrows sketching flow directions. One is coincident with the cross-section, and shows a flow-through river (Hengshui) and a gaining one (Wei). However, there is other arrow indicating that the flow is more or less parallel to this last river. This issue of groundwater flow must be clarified, particularly its pattern in the half of the aquifer to the east of Qishui river, and particularly the relationship of this river with the aquifer: what kind of hydrogeological boundary the river is? is there a flow-through situation associated with it, too? Maybe this can be clarified with an additional cross section.

Later on in the text, when the hydrochemical results are presented, the graphical representations only consider two spatial options: west and east of the abovementioned river, suggesting (without stating it clearly) a general flow from west to east, that is, passing through/below the Qishui (more or less the same as expressed in the cross-section for the Hengshui case). Additionally, there is a lack of information on the groundwater flow scheme in the sector of confluence between the Jing and Wei rivers, where a concentration of sampling points exist.

Taking into account that the general W-E flow path can extend over a length of some 100 km, crossing more than one stream, some mention about the possibility of having local/regional flow systems can be pertinent.

To sum up this comments arising from the examination of the Fig. 1, both the description of the hydrogeological background in the text, and its graphical illustration in Fig. 1, must be modified in order to give more detailed information, but at the same time, trying to make the Fig. 1 (particularly the cross-section, maybe adding other more), easier to visualize in order to extract the basic facts of the groundwater flow.

113        “The aquifer mainly receives atmospheric precipitation and lateral runoff recharge in the piedmont.” Is the irrigation return flow considered as a source of recharge? If so, has it been estimated (as a percentage of the applied volume, for example). Furthermore: What about recharge from rivers? The cross section in Fig. 1 suggests that this can be possible in the two rivers represented to the east of the main one.

114        …” main processes and activities reducing groundwater levels are artificial mining…” (I don’t understand what “artificial mining” means “: does a “natural” mining exist? Or do you mean overpumping?)

122        “a total of 223 groundwater samples were collected in the study area”. I haven’t counted them with detail in Fig. 1, but it seems that the number of red dots is smaller. Is this because some points are nearby and this makes it not possible to differentiate, or it is because there are more than one sample in certain points? By the way, what are the dates of sampling?

128        “…Physico-chemical”. (Why capital letter? )

Delete the final % sign in formula (1)

169        “Total dissolved solids (TDS) concentration was 202.70 mg/L–1,860.00 mg/L, with an average of 675.46 mg/L”. I suggest avoid decimal figures. Better rounded values in this general description.

Table 1. In order to keep some consistency with what is then indicated in the hydrochemical graphs, I recommend including to “average compositions”, corresponding to the two groups of waters represented in those graphs.

193       “Trends in water chemistry in the direction of groundwater flow showed that…” (this seems an oversimplification; see my comments on the Fig. 1).

209         “Figure. 2b shows” (delete point)

Table 2   (STD must be TDS. EC values better without two decimals. 220 or 223 samples, as previously told?

244        “as the terrain becomes lower, groundwater runoff intensity decreases and the leaching effect weakens.” This statement is unclear to me: Does the leaching effect refer to the precipitation+irrigation return-flow? What the term groundwater runoff intensity means, and how do you know it?

Table 3  EC and TSD are variables usually well correlated, and may has no sense include both in an analysis of this type, but in any case the values shown in their two columns are the same, which indicates a transcription error.

342        “in Figureure 5c” (must be Figure)

In the 5.2.1 section there are hypothesis on mineral dissolution/precipitation processes, but no mention is provided on saturation states of the waters concerning the min minerals. This is a drawback of the study. Furthermore, in 77 it is stated “geochemical simulations” as one of the methods used in this study: I wonder how this can be done without estimating the saturation states, and then ¿why the objectives and results of such “simulations” are not mentioned in this study?

Another drawback of the hydrochemical study stems from the lack of information about the composition of the river water, provided that it is supposed to recharge the aquifer to some extent (by irrigation returns or by losing streams). However, this is indirectly addressed in the text when it is stated that in some sectors the contamination that affect the rivers may condition the quality of the water of the aquifer.

Fig. 5     Xing Ping and Xian Yang seems to be some 20 km apart in this map, but approximately half of this value from the map in Fig. 1 (check the scales)

405        “As shown in Figure. 6, by 2119,” (delete dot before 6) (is 2119 OK, or it is 2019? In any case, the X axis in that figure doesn`t allow to identify specific years)

420        “and determine its suitability for domestic use…” (I have hardly noticed in this study this kind of objective; it shouldn`t be mentioned at level of conclusion of this study)

423        “Continuous input from human production and agricultural activities was also found to affect groundwater quality.” (I think that the human effects on groundwater quality have been treated in a very general way; again, it shouldn`t be mentioned at level of conclusion of this study)

426         “Soil water is absorbed by plants through the rhizomes and water is lost in the form of evapotranspiration and evaporation. During evaporation, salt in soil water is retained in the soil, leading to a cumulation of soil salts.” (these statements are not conclusions of this study; remove them)

435        “Along the direction of groundwater flow,” (it seems more consequent/honest to write “From west to east”, because nothing has been detailed about flow directions)

447        “whereas the impacts of human activities are not significant.” ((again, I consider that the human effects on groundwater quality have been treated in a very general way; it shouldn`t be mentioned at level of conclusion of this study)

453         “Even under a scenario of groundwater quality remaining unchanged for a long time, the cumulative loss of total crop output will reach 49.17 tons by 2119, with a cumulative economic loss of 80.817 million yuan.” (maybe I am wrong, but- as I expressed in my first comment- these values don`t seem significant to me for a period of a century; maybe is a defect in expressing them)

Author Response

(The authors gave the same response as above.)

Round 3

Reviewer 1 Report

Fix the unit of EC. It should be microsiemens per centimeter.  Check notation.

Author Response

Response:It has been changed to µS / cm.

Reviewer 2 Report

COMMENTS ON THE 2ND VERSION OF THE WATER 769333 MANUSCRIPT

The authors are considered most of my suggestions to the first version of the MS. However, there are still things that to my mind have not been conveniently addressed or that still remain unclear to me, as I will explain in this second comments.

In my comments to the first version I suggested simplifying, as far as possible, the cross section that accompanied the map in Fig. 1 and clarifying some issues of its legend. Now I see that the cross section has been eliminated. I consider that this implies a reduction on the hydro-geological information of the study area. Please, consider keeping the original structure with the improvements done in the map, and, if possible, drawing the cross section in accordance with them.

…”and has an area of 2,355 km2. The northern part of the study area falls within the piedmont alluvial fan and loess terrace area, whereas the southern region is located in the Wei River terrace. This area contains important grain farms and is also the industrial base in Shaanxi Province. In the study area, the total cultivated land area is 19.54×10^4 hm2 , of which the area that can be irrigated by water is 18.90×10^4 hm2 “ . (it is better to keep the same units as above: 1954 km2 and 1890 km2)

“ Table 1. Physicochemical groundwater characteristics”: OK, but in “Table 3. Matrix plot of the geochemical parameter”. (Change “geochemical parameter” by “Physicochemical groundwater characteristics”. Just to be consistent.)

Table 3: as I told in my previous comments, I consider rather curious that two different physical-chemical variables as CE and TDS, which are not 100 % linearly correlated, show exactly similar correlation coefficients with the other seven variables. Have the authors an explanation for this?

I must say that I am not an English-native speaker, but I wonder if the text has been reviewed concerning the language. Overall, it looks easily readable to me, although in some parts, as the caption of Fig. 6, for example, I have doubts about its correctness.

In the Conclusions section: “Using 2018 as an example, under a scenario of irrigation of wheat and corn with groundwater in the study area, there would be a reduction in annual crop output by 0.489 tons with an economic loss of 0.741 million yuan…Even under a scenario of groundwater quality remaining unchanged for a long time, the cumulative loss of total crop output will reach 49.17 tons by 2119, with a cumulative economic loss of 80.817 million yuan. For many farmers, this is the fundamental to sustain life, and for the city, many people will not be able to eat enough.”(Perhaps I have missed some information to understand the results, both for 2018 and for their extension to a 100 year period: 49 tons of cereals, which can be easily transported by two trucks, are supposed to be the cumulative loss during a century in the study area, costing more than 10 million USD? Please, check the figures. This seems a nonsense. Delete the trivial, subjective last sentence, too.

Author Response

(The authors gave the same response as above.)
